# Public economic gains from tax-financed investments in childhood immunization in the United States

**Mark P. Connolly**[1,2ᴼ]*, **Nikolaos Kotsopoulos**[1,3ᴼ], **Craig Roberts**[4ᴼ],
**Laurence Kotlikoff**[5ᴼ], **David E. Bloom**[6ᴼ], **Tianyan Hu**[4ᴼ], **Mawuli Nyaku**[4ᴼ]

1 Health Economics, Global Market Access Solutions LLC, Mooresville, North Carolina, United States of America, 2 University Medical Center Groningen, Groningen, Netherlands, 3 Department of Economics, University of Athens, Athens, Greece, 4 Merck & Co., Inc., Center for Observational and Real-World Evidence, Kenilworth, New Jersey, United States of America, 5 Department of Economics, Boston University, Boston, Massachusetts, United States of America, 6 Harvard T.H. Chan School of Public Health, Boston, Massachusetts, United States of America

ᴼ These authors contributed equally to this work.
* marconaus@hotmail.com, m.connolly@rug.nl

**Data Availability Statement:** The data on which this analysis were based have been obtained from published sources that are publicly available and freely accessible. All input parameters have been

## Abstract

The emergence of COVID-19 has displayed the importance of immunization and the need for continued public investment in vaccination programs. Globally, national vaccination programs rely heavily on tax-financed expenditure, requiring upfront investments and ongoing financial commitments. To evaluate annual public investments, we conducted a fiscal analysis that quantifies the public economic consequences to government in the United States attributable to childhood vaccination. To estimate the change in net government revenue, we developed a decision-analytic model that quantifies lifetime tax revenues and transfers based on changes in morbidity and mortality arising from vaccination of the 2017 U.S. birth cohort. Reductions in deaths and comorbid conditions attributed to pediatric vaccines were used to derive gross lifetime earnings gains, tax revenue gains attributed to averted morbidity and mortality avoided, disability transfer cost savings, and averted special education costs associated with each vaccine. Our analysis indicates a fiscal dividend of $41.7 billion from vaccinating this cohort. The bulk of this gain for government reflects avoiding the loss of $30.6 billion in present-value tax revenues. All pediatric vaccines raise tax revenues by reducing vaccine-preventable morbidity and mortality in amounts ranging from $7.3 million (hepatitis A) to $20.3 billion (diphtheria) over the life course. Based on public investments in pediatric vaccines, a benefit-cost ratio of 17.8 was calculated for each dollar invested in childhood immunization. The public economic yield attributed to childhood vaccination in the U.S. is significant from a government perspective, providing fiscal justification for ongoing investment.

cited in the manuscript and Supplemental Material with links provided to relevant data sets. No primary patient level data has been used in the conduct of this analysis. No proprietary data has been used for this modeling exercise.

**Funding:** The work reported here was sponsored by Merck Sharp & Dohme Corp., a subsidiary of Merck & Co., Inc., Kenilworth, NJ, USA. The authors maintained full editorial control over content and final publication. No commercial names are used in the analysis described here. The funders had no role in study design, data collection and analysis, decision to publish, or preparation of the manuscript.

**Competing interests:** At the time this work was conducted, CR, TH, and MN were employees of Merck Sharp & Dohme Corp., a subsidiary of Merck & Co., Inc., Kenilworth, NJ, USA and were shareholders of Merck & Co., Inc., Kenilworth, NJ, USA. MPC, NK, LK and DEB received consulting fees in relation to their contributions. MPC, NK, LK, and DEB hold no financial interests in the sponsoring organization. The authors retained full editorial control over the final content. There are no patents, products in development or marketed products associated with this research to declare. This does not alter our adherence to PLOS policies on sharing data and materials.

## Introduction

The bidirectional relationship between health and wealth is one of the few accepted relationships that bonds and transcends the disciplines of economics and public health [1]. The health/wealth relationship is particularly pronounced in infectious diseases with high disease-transmission rates, where mitigation can reduce both short- and long-term demographic/economic damage. Such damage may involve increased mortality, morbidity, and disability; the closure of schools and consequent arrest of human capital development; and/or unemployment, bankruptcies, and disruption of domestic and foreign trade [1]. Central to this is the impact that infectious diseases can have on governments in terms of increased expenditure and lost tax revenues [2]. Thus, government plays a central role in creating development incentives, purchasing and distributing vaccines to combat infectious disease.

Every year the US federal government invests in the Vaccines For Children (VFC) program, purchasing more than half of vaccines used in children [3]. To inform the debate over regular negotiations regarding public vaccination funding, the present analysis measures the annual net government fiscal dividend from ongoing investments in childhood vaccination using an established public economic framework [4–6]. The fiscal framework extends the human capital approach to assess the economic impact of morbidity and mortality attributable to vaccine preventable conditions in children from the perspective of government [2,4,5]. Within our framework we compare how infectious diseases influence government cash flows in terms of lost tax revenue attributed to labor productivity, and changes in transfer payments over the lifetime of the birth cohort with and without immunization. Findings from this analysis provide vital information to government officials operating under the fiscal constraints.

## Materials and methods

### Modeling approach

The fiscal analysis applied a decision analytic model previously developed to estimate the health economic impact of routine childhood vaccination programs in the United States [7,8]. We include 14 vaccine-preventable diseases covered by routine childhood immunization in children aged 0 to 10 years: diphtheria, invasive *Haemophilus influenzae* type b, hepatitis A, hepatitis B, influenza, measles, mumps, pertussis, pneumococcal disease, polio, rotavirus, rubella, tetanus, and varicella [7,8]. The fiscal framework captures rates of vaccination, infection, death, and complications for each vaccine-preventable disease covered by the childhood vaccination program following the ISPOR Good Practices for Outcomes Research recommendations for Economic Analysis of Vaccination Programs [5]. Specifically, we follow the 2017 U.S. birth cohort (3,855,000 births) from birth through death, estimating the epidemiologic and economic outcomes with and without vaccination from the government finance perspective. The framework evaluates changes in morbidity and mortality following the pediatric vaccination schedule and resulting fiscal consequences attributed to excess vaccine preventable events from a previously reported study [8]. The framework compares the costs of public investments in vaccine acquisition and adverse event management in with the cross-sectorial consequences of this public investment on future taxes and transfers. The applied analytic framework is incremental and hence the costs of implementing this publicly funded pediatric vaccination schedule are compared with the expected tax revenue and transfers over the lifetime of a cohort with and without the under - study vaccination schedule. The analysis described here is a modeling study derived from previously published reports, therefore ethics approval was not required for conducting this study.

The fiscal framework captures long-term complications/sequelae associated with several infectious conditions. Complications selected for inclusion were based on a previous analysis by Zhou et al. (2014) [9] and were informed by the published literature. Model inputs related to long-term complications/sequelae included the percentage of cases developing each long-term complication, annual direct costs associated with each long-term complication, the duration of each long-term complication, and complication-related death rates. Long-term complication rates are critical for the fiscal analysis as the fiscal consequences of each disease depend on the incidence of long-term disability in the modeled cohort. Long-term disability results in tax revenue losses due to reduced employment, lower wages, and increased government transfers. These effects are added to the fiscal effect of premature mortality.

## Fiscal analysis

The long-term outcomes of each infectious condition were translated to fiscal costs accrued or averted to quantify the broader economic deficit or surplus produced by public investments in childhood vaccination [2,4,5]. Based on the published literature, the following long-term health outcomes have fiscally relevant impacts:

1. Cognitive impairment and learning disabilities

2. Long-term cognitive and learning disability resulting from encephalitis

3. Disabilities associated with advanced hepatic disease

4. Hearing loss

5. Permanent paralysis.

6. Mortality from vaccine-preventable infections.

We conducted a targeted literature review in November 2020 in PubMed and Google Scholar to identify employment data for people with disabilities that can be associated with vaccine-preventable diseases. The aim of the literature search was to identify previous studies reporting the relative impact of these long-term outcomes on labor market outcomes compared with the general population. Several relevant studies were identified and selected based on completeness of outcomes, study design, and coverage of the U.S. population [10–24]. From the literature search findings, we applied the relative measures of impact for each long-term outcomes e.g., hearing loss, permanent paralysis and cognitive impairment, and its impact on future employment and earnings losses in subjects with these conditions based on outcomes reported by Winsor (2019), Emmett and Frances (2015), and Newman (2011) in the vaccinated and unvaccinated cohorts [12,18,23] (S1 Table). These labor market reductions were translated into lost fiscal revenue for the government.

Additionally, the fiscal outcomes for each vaccine preventable condition and long-term sequalae were associated with various transfer payments due to disability. The underlying assumption applied in the model was that children with permanent disabilities attributed to infectious conditions would have increased dependency on public benefits due to reduced labor market activity. For example, disability in children was associated with higher lifetime Social Security Disability Insurance (SSDI) payments and increased educational costs arising from cognitive impairments. In adulthood, disability was associated with higher lifetime transfer costs from SSDI due to lower levels of employment, and reduced lifetime earnings (S1 Text). Moreover, the longevity gains from vaccinations raise total expected lifetime wages and lifetime tax revenues. Finally, our analysis considers the fiscal effects of vaccine-induced longevity, i.e., healthcare costs and pension costs arising from increases in life expectancy (S2 Text).

## Calculations

Reductions in deaths and comorbid conditions attributed to pediatric vaccines were used to derive gross lifetime earnings gains, tax revenue gains attributed to averted morbidity and mortality avoided, disability transfer cost savings, and averted special education costs associated with vaccinated and unvaccinated cohort. Deaths averted were estimated based on the percentage of incident cases resulting in death for each vaccine-preventable disease under study [7]. All-cause mortality was modelled based on life tables for the United States [25]. To account for future government obligations due to improved survival, we estimated the fiscal consequences of longevity, e.g., Social Security and Medicare after age 65.

Age-specific mean wages on which tax revenue losses were calculated were obtained from the U.S. Census Bureau [26] (S2 and S3 Tables). Consistent with the generational accounting methodology [6], wages were inflated to reflect real wage growth over the duration of working years and adjusted for labor force participation rate by age obtained from the U.S. Bureau of Labor Statistics [27]. All costs and wages were discounted at 3% over the lifetime of the vaccinated and unvaccinated cohorts [28] (S4 Text).

The tax revenue consisted of both direct and indirect tax levies on individuals. Direct taxes were estimated based on the average national tax burden, in addition to income taxes on wages and payroll taxes that are used to fund Social Security, Medicare and Medicaid [29]. Payroll tax contributions from the employer of 7.65% and the employee 7.65% were both applied to annual wages of employed individuals. Additionally, each state levies a state income tax and sales tax in which we applied the US average tax burden as reported by the Tax Foundation [30]. The lifetime taxes were estimated annually and discounted every year in the model (S3 Text).

To estimate the fiscal consequences from investment in pediatric vaccines, we generated fiscal benefit-cost ratios (fBCRs) based on changes in costs or revenue for government from reducing excess vaccine-preventable morbidity and mortality relative to public sector vaccination costs. For the analysis, fiscal savings include those from averted tax losses, reduced disability costs, special education costs and disease related healthcare costs; longevity costs include future unrelated healthcare costs and pension costs associated with survival. We also included unrelated healthcare costs for those individuals receiving tax-financed healthcare i.e. Medicaid. The details of these calculations are described in the supporting information (S4 Table). Additionally, to contextualize our findings, we divided fiscal gains from childhood vaccination without and with longevity costs by nominal gross domestic product (GDP) in the base year to illustrate the percentage contribution attributed to pediatric vaccination [31].

To test model's sensitivity, we applied plausible changes to the discount rate, inflationary measures i.e., CPI, wage growth rates, and vaccine acquisition costs to evaluate the sensitivity of the base case results on the fBCR.

## Results

Total discounted lifetime fiscal benefits attributed to pediatric vaccination within a single birth cohort is estimate at $41.7 billion. This fiscal windfall arises from: (A) averted tax revenue losses $30.6 billion, (B) savings on disability costs $1.6 billion, (C) reduced special education costs $0.91 billion and (D) public-sector healthcare costs $8.6 billion after deduction of publicly-purchased vaccination costs (Table 1, columns A+B+C+D–vaccination costs).

All pediatric vaccines raise tax revenues by reducing vaccine-preventable morbidity and mortality in amounts ranging from $7.3 million (hepatitis A) to $20.3 billion (diphtheria) over the lifetime of the birth cohort. Fiscal savings were also due to averted disability transfer payments and special education costs for children experiencing permanent disablement. Savings

**Table 1. Summary of fiscal effects attributed to vaccination discounted by 3%.**

| Vaccination | Societal perspective | Fiscal perspective | | | | | |
| --- | --- | --- | --- | --- | --- | --- | --- |
| | | Averted tax revenue loss and public costs | | | | Longevity costs | |
| | Gross earnings gain (averted losses) | Averted tax revenues loss (A) | Averted disability transfers cost (B) | Averted special education costs (C) | Disease-related healthcare savings (D) | Unrelated healthcare costs, i.e., Medicare, age >64 (E) | Retirement pensions' costs (F) |
| Diphtheria | $51,487,677,020 | $20,337,632,423 | $0 | $0 | $1,311,795,957 | $2,270,814,556 | $2,273,204,036 |
| Tetanus | $205,463,921 | $81,158,249 | $0 | $0 | $16,895,255 | $10,927,163 | $14,515,694 |
| Pertussis | $1,680,829,731 | $663,927,744 | $0 | $0 | $517,451,245 | $75,885,314 | $78,589,852 |
| Hep A | $18,428,226 | $7,279,149 | $3,551,016 | $0 | $10,770,177 | $886,088 | $1,406,260 |
| Hep B | $1,077,926,784 | $425,781,080 | $149,460,172 | $0 | $43,049,763 | $52,478,266 | $80,647,338 |
| Hib | $3,117,972,431 | $1,231,599,110 | $343,385,450 | $334,038,657 | $1,730,654,856 | $62,494,479 | $60,038,347 |
| Influenza | $278,128,135 | $109,860,613 | $0 | $0 | $218,365,619 | $12,437,585 | $12,279,482 |
| Measles | $5,787,693,939 | $2,286,139,106 | $89,873,435 | $0 | $1,711,295,315 | $242,190,148 | $249,786,068 |
| Mumps | $21,344,224 | $8,430,968 | $0 | $0 | $555,560,966 | $913,560 | $948,378 |
| Rubella | $175,383,299 | $69,276,403 | $0 | $43,355,334 | $113,120,082 | $2,188,164 | $2,201,446 |
| Pneumococcal | $10,800,963,562 | $4,266,380,607 | $759,444,393 | $531,708,222 | $1,747,663,709 | $1,095,514,993 | $1,776,883,220 |
| Polio | $2,518,417,629 | $994,774,963 | $233,897,501 | $0 | $411,322,866 | $37,857,146 | $50,675,357 |
| Varicella | $177,579,833 | $70,144,034 | $19,384,543 | $0 | $119,849,275 | $5,950,884 | $7,006,031 |
| Rotavirus | $32,972,338 | $13,024,073 | $0 | $0 | $137,353,476 | $1,518,806 | $1,455,744 |
| **Total** | **$77,380,781,071** | **$30,565,408,523** | **$1,598,996,509** | **$909,102,214** | **$8,645,148,559** | **$3,872,057,154** | **$4,609,637,254** |

from disability pensions were attributable to six of the infectious conditions with most of these averted costs attributed to hepatitis B, Hib infections, and measles, totaling $149.5 million, $343.4 million, and $89.9 million, respectively.

The inclusion of future longevity costs has a limited impact on the total fiscal benefits for government from childhood vaccination. Specifically, longevity increases government obligations for Social Security benefits and Medicare, which reduces fiscal gains by $billion. This breaks down to increased Social Security benefits of $4.6 billion and Medicare costs of $3.9 billion that are deducted from fiscal gains (Table 1, columns A+B+C+D−E−F−vaccination costs).

The fiscal BCR based on changes in tax receipts and transfer costs relative to public vaccination costs was 17.8 excluding longevity costs. Including longevity costs attributed to Social Security and non-related healthcare obligations decreased the fBCR to 3.9.

The estimated economic gains from childhood vaccination as a proportion of GDP from the net fiscal gains including longevity from a single birth cohort without and with longevity costs represent 0.18% and 0.14% of GDP in 2019, respectively.

The results of the scenario analysis show high sensitivity to the selection of discounting rate with the scenario of 0% discount rate resulting in high fBCRs due to the impact of productivity growth (Table 2). Including the costs attributed to longevity increased the fBCR to 1.6 and 8.1 for discount rate set to 0% and 5% wage increases, respectively. By comparison, our findings indicate that changing vaccine costs had limited impact on the fBCR when evaluated with or without the effects of longevity included. A scenario that set wage growth and inflationary measures to 0% and zero discount rate lowered the fBCR to 1.2 when effects of longevity were included.

## Discussion

Public expenditure on childhood vaccination in the United States (US) includes purchases through the VFC, with additional contributions from Section 317 and state and local

**Table 2. Scenario analysis of core model inputs and impact on fiscal benefits cost ratio with and without longevity expenditure.**

| Fiscal BCRs | With longevity effect | Without longevity effect |
|---|---|---|
| Scenario 1: -10% vaccination acquisition cost | 3.9 | 19.7 |
| Scenario 2: +10% vaccination acquisition cost | 3.8 | 16.2 |
| Scenario 3: Discount rate 5% | 5.0 | 9.6 |
| Scenario 4: Discount rate 0% | 1.6 | 64.2 |
| Scenario 5: CPI rate 0% | 7.2 | 17.4 |
| Scenario 6: CPI rate 3% | 1.3 | 18.5 |
| Scenario 7: Wage growth 0% | 1.7 | 8.1 |
| Scenario 8: Wage growth 5% | 8.1 | 37.5 |
| Scenario 9: Discount rate 0%, CPI rate 0%, Wage growth 0% | 1.2 | 17.9 |
| **Base case Scenario** | 3.9 | 17.8 |

purchases, which together purchase more than half of vaccines used in children [3,32]. Under the terms of the VFC, contracts are negotiated between manufacturers and the Secretary for defining vaccine purchases and delivery [33]. Both VFC and Section 317 have shown to increase vaccination coverage, and VFC has been shown to reduce income disparities in vaccination uptake. High coverage levels enabled by public-sector support for vaccination contributes to maintenance of herd protection and sustained low rates of vaccine-preventable diseases in the US population. However, the economic value of public financing for immunization has not been studied, and the sustainability of these programs has been questioned [32]. Additionally, with the introduction of the Inflation Reduction Act by Congress in 2022, the role of economic data could be increasingly important for establishing value of technologies, although the law stops short on the role of defining economic value from changes in outcomes [34].

In support of public spending on vaccination, this paper reveals the significant government benefits associated with public sector investment in the U.S. childhood immunization program that derive from reductions in morbidity and mortality associated with 14 childhood disease vaccines recommended by the Advisory Committee on Immunization Practices. We estimate $30.6 billion in tax revenue losses, $1.6 billion in disability transfer payments, and $909 million in special education cost can be averted from childhood vaccination over the lifetime of the 2017 birth cohort, representing discounted benefits exclusively gained by the government. Furthermore, we estimate that $1 spent on vaccines generates $17.8 in fiscal return over the lifetime of the birth cohort.

The analysis reported here considers a birth cohort that includes disabled and non-disabled individuals and those that die from infectious conditions for which we quantify lifetime fiscal losses. The lifetime present value of tax revenue loss due to mortality is estimated based on the age of death. Similarly, the model calculates the percentage loss of earnings and tax revenue due to disability and cognitive disability. Depending on the age of disability the present value of tax revenue loss is estimated for the remaining life expectancy of the disabled person. Averted deaths and disability cases are then multiplied by the averted tax revenue loss. On average, the government benefits from each incremental life that is saved in terms of future tax revenue. However, in reality the government benefits substantially more from preventing disability as these individuals require a lifetime of income support and will pay fewer taxes due to reduced work activity. For example, the disability costs paid by Social Security for an individual every year are more than $14,000 annually [35]. Although permanent disablement is rare

for pediatric infectious diseases, for completeness we included these cost estimates in our fiscal projections.

Vaccine-induced longevity costs were approximately $4.6 billion in retirement pension costs and $3.8 billion future Medicare costs. In a scenario where vaccines were absent, this would represent a potential fiscal loss of 0.18% in GDP based on the lost lifetime earnings of a single birth cohort.

Findings from the univariate sensitivity analysis suggest that the discount rate applied is one of the most impactful parameters on government cash flows. While there are some suggestions to use differential discount rates or lower discount rates for longer time horizons [5], we have applied a single discount rate of 3% over the duration of the model which suggests our findings might be conservative. When forecasting government revenues, standard practice by federal agencies is to discount future cash flows using projected yields on Treasury securities as this more appropriately reflects the future value of money to the government [36]. As Treasury securities have been at historic lows over the past several years [37], it was felt it would be more conservative to use the accepted 3% discount rate normally used in health economic literature as the base case. As shown in our analysis, varying the discount rate had a significant impact on the fBCR, however in both extreme scenarios the yield on pediatric vaccination remained positive.

In the scenario analysis it was observed that vaccine acquisition costs do not dramatically influence the fiscal yield for government. In our scenario analyses, changing the acquisition cost by +/- 10% resulted in a fBCR of 3.8 and 3.9, respectively when compared to the base case of 3.9 when costs of longevity are included (Table 2). This is attributed to the fact that acquisition costs occur early in the fiscal life and can yield benefits over many years, and the human capital gains–and associated taxes derived from wages–are sustained over many years in comparison with upfront vaccine investment costs. While vaccine costs are important for government in the short-term, over the long-term public economic gains are more than enough to cover small variations in prices.

Public economic assessments seek to reflect the real impact of inflation and tax revenue on public accounts in relation to funding decisions. To reflect these changes, it is necessary to adjust expenditure for inflation, and future wage growth, both of which influence government revenue projections as applied in many types of public economic assessments [6]. As noted in our scenario analysis, future wage growth had a pronounced effect that more than doubled the fBCR when wages grow at 5% increasing the fBCR to 8.1, and when wage growth was zero, the fBCR reduced to 1.7 when the effects of longevity were included. In contrast, decreasing inflation impacts government spending on transfer programs as these are often linked to CPI, therefore decreases government spending relative to revenue. This explains why the fBCR increases to 7.2 (with longevity) in the absence of inflation in the model (Table 2).

The averted fiscal costs described here are largely attributable to changes in mortality due to reductions in vaccine-preventable infectious conditions in children which are both highly contagious and have high mortality rates. The magnitude of fiscal gain varies depending on the number of deaths averted and the timing of such deaths. Diphtheria causes fewer deaths than pneumonia-related infections. However, these deaths occur mostly in childhood, preventing children from ever entering the workforce, therefore depriving government of future tax revenue on potential wages. In contrast, pneumococcal infections occur over the lifetime and mostly concentrated in later years. As many of these individuals have paid taxes early in life, the fiscal loss is less pronounced compared with a condition where mortality is concentrated in children. As few infectious conditions give rise to permanent disability that can influence future education attainment and employment trajectory, the costs of special education and disability represent 6% of averted fiscal costs. By contrast, diseases with high morbidity in infants,

but low mortality or disability consequences, such as rotavirus and varicella, have lower fiscal consequences because the disease experience typically occurs in childhood and seldom leads to death. While preventing these diseases may be cost-effective from a societal perspective, their impacts are less pronounced from a fiscal perspective [7,8].

The public good of protection from disease is achieved in the US through a combination of public and private financing. The contribution of one or the other financing source cannot be disentangled, as herd immunity requires high coverage rates for many diseases. Private health insurers benefit from reductions in vaccine-preventable disease case rates, which reduces hospitalizations and demand for medical services. Similarly, health service cost savings extend to government and taxpayers for those treated through public programs. The VFC program, established in 1993 after successive years with high rates of measles cases, has been shown to increase coverage rates by removing financial barriers to vaccination for a large portion of the population. Since implementation, the US achieved greater than 90% coverage for many routine childhood vaccines [38]. In our analysis, we account for the main fiscal costs and benefits that result from the sustained high level of coverage that public and private financed.

All members of society interact with government public accounts through taxation and redistribution of income through social benefits, including public health programs. Within this social contract, all members of society have a vested interest in the health of their neighbors and in ensuring they remain healthy and productive, accumulate wealth, and continue paying taxes. Moreover, what is not captured in this analysis is the deadweight loss associated with premature death or disablement as remaining workers or future workers face higher taxes to pay for social benefit programs promised to others. In this context, all members of society benefit from each other's commitment to vaccinate, thereby reducing deadweight losses that can lead to increased taxes, and increased prices for all. The fiscal analysis presented here spotlights these relationships and demonstrates the accrued benefits from investing in vaccination programs and the benefits of reducing disease transmission of vaccine-preventable conditions.

## Conclusions

There are many justifications for investing in public vaccination programs for preventing childhood infectious conditions. In the fiscal analysis described here we provide further justification on fiscal grounds and the resulting tax revenue gains and reduced transfers associated with preventing infectious conditions.

## Supporting information

**S1 Table. Clinical study reporting outcomes linking to fiscal consequences.**
(DOCX)

**S2 Table. U.S. Census Bureau median and mean income from Current Population Survey.**
(DOCX)

**S3 Table. Labor force participation.**
(DOCX)

**S4 Table. Nonrelated healthcare (Medicaid) costs.**
(DOCX)

**S1 Text. Disability costs and special education costs.**
(DOCX)

**S2 Text. Costs of longevity paid by government.**
(DOCX)

**S3 Text. Estimation of lifetime direct and indirect taxes.**
(DOCX)

**S4 Text. Cost inflation, discounting, and wage growth.**
(DOCX)

## Author Contributions

**Conceptualization:** Mark P. Connolly, Nikolaos Kotsopoulos, Craig Roberts, Laurence Kotlikoff, David E. Bloom, Tianyan Hu, Mawuli Nyaku.

**Data curation:** Mark P. Connolly, Nikolaos Kotsopoulos, Laurence Kotlikoff, David E. Bloom, Mawuli Nyaku.

**Formal analysis:** Mark P. Connolly, Nikolaos Kotsopoulos, Craig Roberts, Laurence Kotlikoff, David E. Bloom, Tianyan Hu, Mawuli Nyaku.

**Funding acquisition:** Mark P. Connolly, Craig Roberts, Laurence Kotlikoff, David E. Bloom, Mawuli Nyaku.

**Investigation:** Mark P. Connolly, Nikolaos Kotsopoulos, Craig Roberts, Laurence Kotlikoff, David E. Bloom, Tianyan Hu, Mawuli Nyaku.

**Methodology:** Mark P. Connolly, Nikolaos Kotsopoulos, Craig Roberts, Laurence Kotlikoff, Tianyan Hu, Mawuli Nyaku.

**Project administration:** Mark P. Connolly, Craig Roberts, Mawuli Nyaku.

**Supervision:** Mark P. Connolly, Craig Roberts, Laurence Kotlikoff, Tianyan Hu, Mawuli Nyaku.

**Writing – original draft:** Mark P. Connolly, Nikolaos Kotsopoulos, Laurence Kotlikoff, David E. Bloom, Tianyan Hu, Mawuli Nyaku.

**Writing – review & editing:** Mark P. Connolly, Nikolaos Kotsopoulos, Craig Roberts, Laurence Kotlikoff, David E. Bloom, Tianyan Hu, Mawuli Nyaku.

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
