## [Decision Letter · Decision Letter 0]

26 Jun 2023

PGPH-D-23-00704

Public economic gains from tax-financed investments in childhood immunization in the United States

Dear Dr. Connolly,

Thank you for submitting your manuscript to PLOS Global Public Health. After careful consideration, we feel that it has merit but does not fully meet PLOS Global Public Health’s publication criteria as it currently stands. Therefore, we invite you to submit a revised version of the manuscript that addresses the points raised during the review process.

We look forward to receiving your revised manuscript.

Kind regards,

Susmita Chatterjee, Ph. D

Academic Editor

Journal Requirements:

1. Please send a completed 'Competing Interests' statement, including any COIs declared by your co-authors. If you have no competing interests to declare, please state "The authors have declared that no competing interests exist". Otherwise please declare all competing interests beginning with the statement "I have read the journal's policy and the authors of this manuscript have the following competing interests:"

Additional Editor Comments (if provided):

Reviewers' comments:

Reviewer's Responses to Questions

**Comments to the Author**

1. Does this manuscript meet PLOS Global Public Health’s publication criteria? Is the manuscript technically sound, and do the data support the conclusions? The manuscript must describe methodologically and ethically rigorous research with conclusions that are appropriately drawn based on the data presented.

Reviewer #1: Yes

Reviewer #2: Yes

2. Has the statistical analysis been performed appropriately and rigorously?

Reviewer #1: Yes

Reviewer #2: Yes

3. Have the authors made all data underlying the findings in their manuscript fully available (please refer to the Data Availability Statement at the start of the manuscript PDF file)?

Reviewer #1: Yes

Reviewer #2: Yes

4. Is the manuscript presented in an intelligible fashion and written in standard English?

Reviewer #1: Yes

Reviewer #2: Yes

5. Review Comments to the Author

Reviewer #1: REVIEW

PGPH-D-23-00704

Public economic gains from tax-financed investments in childhood immunization in the United States

Thank you for asking me to review the paper. The paper estimates the monetary benefits of deaths and disabilities averted from vaccine that prevents several illnesses that can occur anytime from a very early age (although some only during childhood). It calculates lifetime wages from workers who are statistically alive or not disabled due to the vaccine interventions. It also calculates net contributions to the US fiscal balance: taxes minus longevity entitlements and disability transfers. It compares the present value of the estimate to the vaccine costs. This is a very worthwhile exercise and has been done accurately using currently available data. The paper deserves to be published. This exercise is worth carrying out and should be presented to Congress and other governance bodies.

I don’t have much to say; I will make only a few points.

1. It would be nice to get some basic numbers with an example. This would give the general picture of your assumptions and some cogency to the numbers. Diphtheria, for example, has a case-fatality rate of 5-10% but could be 20% for anyone < 5 years of age and > 40 years of age. How did you pick the number of years lived when a death is avoided. What are the assumptions of the base wage for example. This is just to get a handle on this for the reader when they read it. Some of this is in the supplementary material.

2. My own exercise at the median wage estimated that a person’s tax contribution to social security and Medicare transfer has a ratio around 2.4 (PV $190,000 tax to $-90,000 transfer). This roughly is about 41% amount going to longevity transfer; I am assuming cost of education and other things are covered from the $190k. The amount 41% might be slightly an overestimate from reality; but not too far off. But averting a death amounts to a net gain of $100,000 present value. What is the corresponding value for averting a disability? An estimate: Suppose a person becomes disabled at the age of 21 and lives 30 years more and government pays net transfer of $50000 per year for care. The care has a present value of $550,000 as costs. Suppose the question is to avert a death or a disability at the margin. Disability always wins out for the government. This is not to say the approach taken here is at all wrong. It would be nice to simply to say that this methodology at this time is intended only to show that vaccine programs with most plausible assumptions are likely to yield a great deal of benefit. However, making decisions on human lives based on net fiscal benefits is not a good idea. It is just to note some limitations.

Reviewer #2: The paper studies the value of childhood vaccinations from the perspective of the US government. The paper is interesting and written well. It is widely acknowledged, at least in academic circles, that vaccination provides substantial benefits. The rationale the paper provides for studying the value is “to inform debate over regular negotiations regarding public vaccination funding.” To articulate the decision(s) the analysis supports (and how it can be used), it would be useful to provide more background on these negotiations and the debate beyond estimating the value of individual vaccines and comparing their value to decide which should be included in the vaccination programme.

The analysis is thorough and conducted well.

My primary query is about the longevity costs. The unrelated future costs are comparatively low. How do the unrelated future costs of vaccines relative to related costs compare to other studies? In this study, are they just accounted for from >64 yo? I do expect costs to be highest later in life but also suspect these are heavily discounted at this point, being quite far into the future. And, with vaccine-preventable deaths occurring early in life, I think that would have explained the much lower longevity costs. However, scenarios 3 & 4 in Table 2, seem to go in the opposite direction: not discounting yields greater benefits without longevity effect and lower benefits with longevity effect. This suggests that discounting is not the reason. Could it be due to the shorter timeframes over which later-life diseases can the overall value?

I was also wondering how representative the unrelated future costs are for the cohort of newborns in both scenarios – with and without vaccines. Additionally, how do the methods for unrelated future costs compare to methods in the literature on estimating this cost – e.g., van Baal et al. (2011) in Pharmacoeconomics?

6. PLOS authors have the option to publish the peer review history of their article (what does this mean?). If published, this will include your full peer review and any attached files.

**Do you want your identity to be public for this peer review?** For information about this choice, including consent withdrawal, please see our Privacy Policy.

Reviewer #1: **Yes: **Arnab K Acharya, PhD, MPH

Reviewer #2: No

---

## [Editor Report · Decision Letter 1]

15 Sep 2023

Public economic gains from tax-financed investments in childhood immunization in the United States

PGPH-D-23-00704R1

Dear Dr. Connolly,

We are pleased to inform you that your manuscript 'Public economic gains from tax-financed investments in childhood immunization in the United States' has been provisionally accepted for publication in PLOS Global Public Health.

Best regards,

Susmita Chatterjee, Ph. D

Academic Editor